# Oxidative Stress, Antioxidant Capabilities, and Bioavailability: Ellagic Acid or Urolithins?

**DOI:** 10.3390/antiox9080707

**Published:** 2020-08-04

**Authors:** Silvana Alfei, Barbara Marengo, Guendalina Zuccari

**Affiliations:** 1Department of Pharmacy, University of Genoa, Viale Cembrano, 4, I-16148 Genoa, Italy; zuccari@difar.unige.it; 2Department of Experimental Medicine—DIMES, Via Alberti L.B. 2, I-16132 Genoa, Italy; barbara.marengo@unige.it

**Keywords:** oxidative stress (OS), reactive oxygen and nitrogen species, antioxidant effects, ellagitannins (ETs), ellagic acid (EA), urolithins (UROs), human metabotype, pro-oxidant effects, EA-enriched food supplements

## Abstract

Oxidative stress (OS), triggered by overproduction of reactive oxygen and nitrogen species, is the main mechanism responsible for several human diseases. The available one-target drugs often face such illnesses, by softening symptoms without eradicating the cause. Differently, natural polyphenols from fruits and vegetables possess multi-target abilities for counteracting OS, thus representing promising therapeutic alternatives and adjuvants. Although in several in vitro experiments, ellagitannins (ETs), ellagic acid (EA), and its metabolites urolithins (UROs) have shown similar great potential for the treatment of OS-mediated human diseases, only UROs have demonstrated in vivo the ability to reach tissues to a greater extent, thus appearing as the main molecules responsible for beneficial activities. Unfortunately, UROs production depends on individual metabotypes, and the consequent extreme variability limits their potentiality as novel therapeutics, as well as dietary assumption of EA, EA-enriched functional foods, and food supplements. This review focuses on the pathophysiology of OS; on EA and UROs chemical features and on the mechanisms of their antioxidant activity. A discussion on the clinical applicability of the debated UROs in place of EA and on the effectiveness of EA-enriched products is also included.

## 1. Introduction

The health effects of many fruits, fruit juices, nuts, and seeds have been associated with their high content of antioxidant polyphenols and particularly in ellagitannins (ETs) able to provide ellagic acid (EA), one of the most powerful antioxidant molecules [1,2,3]. Food chemists consider both ETs and EA as nutraceuticals, because in addition to possessing the basic nutritional values, they are gifted with several extra health benefits, therefore the dietary intake of foods containing these components often translates in relevant biological effects. For example, documented findings assert a correlation among the consumption of ET-rich foods and greater cardiovascular health [1,2,3,4] or among the consumption of fruits and vegetables and minor incidence of coronary heart disease [5]. Moreover, much empirical data led to the hypothesis that both EA and ETs might be exploited to prevent chronic and degenerative diseases such as cancer, diabetes, cardiovascular diseases, and central nervous system (CNS) disorders [6].

The first evidence concerning the pharmacological properties of some fruits, vegetables, herbs, seeds, and nuts, comes from folk medicine and ethno-pharmacology. In most cases, this knowledge was largely improved because of the interest in research studies focused on the comprehension of the mechanisms and constituents accounting for the therapeutic effects.

Since the late 90s, EA benefits have been reviewed by many authors [1,2,7,8] and both in vitro and in vivo studies have been carried out in an attempt to define the molecular and cellular events rationalizing the biological activities that EA and/or its derivatives exert in pathological conditions. In vitro, EA showed considerable health effects including anti-carcinogenic [1,2,9,10,11,12], anti-atherogenic [13], anti-thrombotic [14,15], anti-angiogenic [16], anti-neurodegenerative [9,17,18,19,20] properties, as well as the capability to prevent obesity [21].

In this context, EA attracted the interest of researchers for its potential health benefits in pathological conditions and nowadays, several literature data support such EA ability [22,23].

The mechanisms at the basis of the EA multifaceted bioactivity relies mainly on its antioxidant and anti-ageing power and on its ability in counteracting the detrimental reactive oxygen and nitrogen species (RONS), which are a byproduct of physiologic aerobic metabolism. In normal conditions, RONS generation is kept under control by cellular antioxidant defenses and repair systems and is involved in the production of energy from organic molecules, in immune defense, and in the signaling process [24]. When overproduced, the detoxification systems of cells fail and RONS accumulate playing an important role in the onset of oxidative stress (OS) and inflammation, in causing irreversible damage to DNA, lipids, and proteins, thus promoting aging, age-related diseases and several degenerative disorders [25].

For more clarity, OS refers to a cascade of events that frequently triggers and accompanies the molecular/cellular pathogenic events, responsible for several human disorders, including carcinogenesis [26,27], atherosclerosis, cardiovascular, and neurodegenerative diseases [28,29]. Differently, inflammation, being both the cause and the effect of RONS accumulation, is considered a pathological hallmark of the most part of human diseases including those that develop in the CNS.

Anyway, the relevance of EA and ETs efficacy is hampered by the low oral and blood-availability of these constituents, following their in vivo administration. Although ETs are water-soluble macromolecules [30], they are not bioavailable and rarely are detected in human plasma or in tissues after normal consumption of ET-rich foods. Many factors dictate for ETs poor bioavailability, such as their large size ranging between 634 and 3740 Da, their relatively high polarity, the presence of a C-C linkage, and their probable binding to some proteins in saliva, that limits their further metabolism and causes astringency [31]. While some ETs are even resistant to acid and basic hydrolysis and can reach the large intestine almost intact [9,32], most are sensitive to acidic and basic pH and are promptly metabolized in stomach and duodenum, respectively, releasing free EA [1,2,9,33].

Concerning EA, although it is a smaller molecule, it has a much lower water solubility than ETs and therefore it is in turn poorly bioavailable [9,34], not absorbable, both in the stomach and in small intestine, and passes unmodified to the grand intestine tract, where it undergoes an extensive metabolism by the gut microbiota providing urolithins (UROs). A recent study investigating the absorption of a standardized extract from pomegranate in healthy human volunteers, after the acute consumption of 800 mg of extract, indicated that EA maximum concentration (C_max_) in plasma was 33.8 ± 12.7 ng/mL at 1 h (t_max_) [35]. In another work about dietary ETs consumption, the amount of EA metabolic derivatives, such as methyl and dimethyl ethers or glucuronic acid conjugates detected in plasma and urine at 1 and 5 h after ingestion, corresponded to very low concentrations as well, not sufficient to produce significant beneficial effects [35,36,37]. A further explanation for EA poor bioavailability relies on the EA aptitude to bind irreversibly to cellular DNA and proteins, or to form poorly soluble complexes with calcium and magnesium ions, which strongly limit transcellular absorption [37,38,39].

Differently, UROs formed for gut microbiota action, since being dibenzopyran-6-one derivatives with different hydroxyl substitutions and a more lipophilic character, are 25–80-fold more bioavailable and are much better absorbed than EA. Indeed, after ET-rich foods intake UROs are the most abundant phenolic compounds detected in blood, urine, and tissues as reported in Section 7 and therefore it was easier to evaluate their in vivo activity [9].

Nowadays, among researchers, there is a widespread tendency to think that UROs, rather than EA, could be the actual bioactive molecules responsible for benefits coming from ETs and EA rich foods [9,36]. This hypothesis is supported by the awareness that, although in vitro findings have shown that EA and UROs are equally active, studies in vivo have been able to provide assessments only with regard to UROs, since only these metabolites have been found in fluids, cells, and tissues.

As for the knowledge acquired so far, EA would exercise its therapeutic activities only by its metabolite UROs.

In order to be able to obtain certainty about EA’s behavior in vivo, scientists have been increasingly and are incessantly focused on preparing water soluble and absorbable EA formulations, able to protect EA and to reduce or nullify EA metabolism to UROs [40]. The formulation of drug delivery systems, capable of transporting and releasing EA to the target site, represents a valid approach for bypassing the bad biopharmaceutical features of the polyphenol, thus allowing a better evaluation of its potential application as radical scavenger antioxidant therapeutic.

In addition, UROs, even if endowed with healthy properties similar to those of EA, are not advisable for safer therapeutic purposes. Actually, UROs are double faceted molecules, able to be beneficial but, depending on their structure, environmental conditions, the cells under study, age, and health state of the individuals, could result also harmful (see details in Section 6) [41,42]. The amount and typology of UROs produced in the gut of individuals depends on the metabolic activity of the microbiota, that is typified by a highly inter-individual heterogeneity [9]. UROs absorption, blood and tissue concentration, and inter-subject variability in the responses to UROs exposure, are unpredictable variables, which lead to heterogeneous comebacks that, paradoxically, could promote adverse effects (see details in Section 6).

In addition, human microbiota activity is difficult to reproduce in animal models, and cannot be easily studied and controlled [38,43,44,45,46,47].

Leaving long and detailed dissertations on the ETs and EA sources and on their health effects to the already existing reviews, the present one focuses preferentially on the physio-pathologic role of OS, on the chemical aspects of EA and UROs, on the mechanisms of their antioxidant effects, and on EA poor solubility and metabolic faith to UROs.

In view of a possible use of EA and/or UROs as template molecules for the development of new antioxidant drugs or as therapeutics, an objective discussion on the lights and shadows of URO systems and on the current trend that considers UROs as the actual substances responsible for beneficial effects deriving from ingestion of ET-rich food, and as more bioavailable alternatives to EA, was provided.

Finally, the author’s critical opinion about the production and administration of functional foods and food supplements enriched with free insoluble EA, to increase its daily intake, *versus* the more rational approach, based on preparing soluble and absorbable EA formulation, has also been included.

## 2. Oxidative Stress

### 2.1. Cell Dysfunctions and Pathological Conditions Associated with Reactive Oxygen Species (ROS) and Nitrogen Species (RNS) Overproduction and OS

Reactive oxygen species (ROS) and nitrogen species (RNS), together RONS, include both radicals and not radical species able to provide radicals in suitable conditions.

These reactive species originate as byproducts of respiration and oxidative metabolism and in small amounts from normal cellular processes. In absence of mitochondrial dysfunction and under physiological conditions, a balance between RONS production and their detoxification by endogenous antioxidant defense mechanisms (enzymes and antioxidant vitamins) ensures controlled levels of RONS, from which cells homeostasis depends [48].

When RONS concentrations overcome the basal and physiological threshold, OS occurs, and a huge increase in blood flow resistance, paralleled by decreased nitric oxide (NO) bioavailability, as well as reduced immune responses happen, which are responsible for cell senescence and aberrant apoptotic events. All these effects, together with oxidative damage to DNA, lipids, and proteins, contribute to cell dysfunction and senescence, and promote aging and age-related pathological conditions.

Cardiovascular diseases (CVDs), chronic obstructive pulmonary diseases (COPDs), chronic kidney diseases (CKDs), neurodegenerative diseases (NDs), macular degeneration (MD), biliary diseases, and cancer are several acute and chronic age-related pathologies due to OS, and show different types of OS biomarkers (Figure 1).

RONS-induced senescent cells are subject to irreversible mutations, that lead to the secretion of soluble factors (interleukins, chemokines, and growth factors), of degradative enzymes like matrix metalloproteases (MMPs), and of insoluble proteins/extracellular matrix (ECM) components, that contribute to determine inflammation, hypertension, and endothelial dysfunction, thus favoring pathological settings.

Moreover, OS is also a key player in cancer development. Consequently, to a metabolic adaptation, cancer cells produce high amounts of ROS, potentially harmful for healthy cells, but simultaneously, they are equipped with increased levels of antioxidants able to efficiently counteract OS and to defend themselves from impairments evoked by OS [27,49].

In fact, in cancer cells, several redox-sensitive transcription factors [protein p53, nuclear erythroid related factor 2 (Nrf2), nuclear factor kappa-B (NFκB)] are hyper-activated and are able to modulate the expression of both antioxidant genes and of signal transduction proteins [protein kinase C (PKC), mitogen-activated protein kinase (MAPK), serine-protein kinase (ATM), etc.].

Through to the constitutive activation of these redox-sensitive pathways, cancer cells ensure themselves the ability to proliferate and reduce death.

To induce age-related pathologies, RONS act on various pathways affecting several cellular processes as reported in Table 1.

### 2.2. RONS Origin

RONS can derive from both endogenous and exogenous sources (Figure 2, Table 2) [50].

Phagocytic cells (neutrophils, monocytes, or macrophages) use NOX for one-electron reduction of molecular oxygen to the radical superoxide anion (O_2_^•^^−^), during cellular respiration. O_2_^•^^−^ is considered the primary ROS, and when reacting with other molecules through enzymatic or non-enzymatic processes, the latter catalyzed by metals, generates secondary ROS.

O_2_^•^^−^ is mainly dismutated by SOD into the hydrogen peroxide (H_2_O_2_), which is able to form the highly reactive ROS hydroxyl ion (^•^OH) and radical HOO^•^, through the Fenton or Haber–Weiss reactions. O_2_^•^^−^ is produced also from the irradiation of molecular oxygen with UV rays, photolysis of water, and by exposure of O_2_ to organic radicals formed in aerobic cells such as NAD^•^, FpH^•^, semiquinone radicals, cation radical pyridinium, or by hemoproteins.

While the radical O_2_^•^^−^ does not react directly with lipids, polypeptides, sugars, or nucleic acids, ^•^OH reacts especially with phospholipids in cell membranes and proteins.

Furthermore, H_2_O_2_ can be converted by chloride and MPO to hypochlorous acid, particularly hazardous for cellular proteins.

As a defense mechanism, by using three different kinds of NOS, i.e., epithelial NOS, neuronal NOS, and inducible NOS, cells generate NO from intracellular *L*-arginine, which is converted to NO^•^ due to NADPH as an electron source. NO^•^ and ONOO^−^ are produced by reaction of NO with ROS, while NO^•^ in combination with O_2_, provides ONOO^•^, which induces lipid peroxidation in lipoproteins [24,51,52].

Table 3 summarizes the most representative radicals and the reactive species ONOOCO_2_^−^ produced in biological aerobic systems.

Whatever their origin, RONS cause indifferently detrimental oxidative modifications of cellular macromolecules such as carbohydrates, lipids, proteins, and DNA, producing molecules, considered also markers of OS (Table 4).

In order to counteract these detrimental effects, cells have developed several repair systems able to repair or eliminate those lipids, proteins, and DNA damaged by the action of RONS. In particular, cytosolic and mitochondrial DNA repair enzymes include polymerases, glycosylases, and nucleases, while proteinases, proteases, and peptidases make up part of the proteolytic enzymes, which take care of removing damaged proteins.

In addition, biological systems have developed both physiological and biochemical mechanisms in order to minimize free radicals (FRs) production and reactive species toxicity. At physiological level, the microvascular system exerts the function of maintaining the levels of O_2_ in the tissues, while at biochemical level a protective activity is exerted both by endogenous (enzymatic and non-enzymatic) and exogenous molecules, as reported in Table 5.

In this regard, GSH-Px, glutathione reductase (GR), and methionine sulfoxide reductase (MSR) act as intermediaries in the repair process of oxidative damage.

## 3. Chemical Insights of EA

### 3.1. EA Chemical Structure and Physical Properties

EA is a chromene-dione derivative whose chemical name is 2, 3, 7, 8-tetrahydroxy-chromeno[5,4,3-cde]chromene-5,10-dione, which can also be seen as a dimeric derivative of gallic acid (GA) (Figure 3) and whose physicochemical properties are summarized in Table 6 [40,58].

EA appears as cream-colored needles (from pyridine) or yellow powder, odorless, and incompatible with strong reducing agents. EA possesses a highly thermostable structure with a melting point of about 450 °C and a boiling point of 796.5 °C [34]. Due to the weak acidic nature of its four phenolic groups (pKa1 = 5.6 at 37 °C), around neutral pH it is mainly deprotonated on positions 8 and 8′, while above pH 9.6 lactone rings open to give a carboxyl derivative [40].

EA structure includes both a hydrophilic moiety composed of four hydroxyl and two lactone groups and a lipophilic planar fragment, consisting of two hydrocarbon phenyl rings. Consequently, EA is equipped with a high degree of crystallinity, deriving from its planar and symmetrical structure and from the extensive hydrogen-bonding network, which can form within the crystals. These structural characteristics, far from being an advantage, lead the acid to be poorly soluble both in aqueous or in organic solvents. EA possesses an insignificant water solubility of about 9.3–9.7 µg/mL at pH 7.4 and 21 °C [8] and a very poor solubility in alcohol [33], which concretize in a very poor bioavailability and trivial absorption in gastrointestinal tract (GIT). EA water solubility increases with pH, as well as the antioxidant action [40]. EA is almost insoluble in acidic media and distilled water, while its water solubility is significantly improved by basic pH. However, in basic solutions, phenolic compounds lack stability and these molecules, under ionic form, undergo extensive transformations or are converted to quinones, as a result of oxidation. A stability study on pomegranate fruit peel extract demonstrated that EA content significantly decreases in a few weeks regardless of the pH of the solution, due to the hydrolysis of the ester group with hexahydroxydiphenic acid formation, suggesting that EA should not be stored in aqueous medium [40].

Concerning organic solvents, EA is slightly soluble in methanol, more soluble in ethanol (EtOH) and dimethyl sulfoxide (DMSO), and shows maximum solubility in *N*-methyl-2-pyrrolidone (NMP), confirming the effect of basic pH on EA dissolution [40].

In this regard, high concentrations of EtOH (80% or greater) could be a suggestion to solubilize EA, but such solutions are not advisable for clinical purposes. Similarly, highly diluted DMSO solutions of EA could be achievable, but DMSO is very harmful to humans. One of the most exploited vehicles for EA is polyethylene glycol (PEG) 400, as it is endowed with satisfactory biocompatibility and, at the same time, is miscible with both aqueous and organic solvents [59]. EA solubility in oils and surfactants is also provided, helpful for developing emulsifying-based techniques [40]. EA poor solubility not only prevents it from reaching cells in vivo, but also causes several difficulties in developing any EA pharmaceutical formulations [60].

### 3.2. EA In Vivo Formation and Metabolism

As already reported in previous sections, except for a marginal amount (e.g., 0.7–4.7 mg/100 g of berries, fresh weight), free form of EA is produced mostly in vivo, essentially upon physiological massive ETs hydrolysis in the stomach and by gut microbiota action in the small intestine. In particular, the ETs hydrolysis leads to the production of GA and hexahydroxydiphenoic acid (HHDP), that spontaneously lactonizes to EA also known as 4,4,’5,5’,6,6’-hexahydroxydiphenic acid 2,6,2’-6-dilactone [1,2,9]. Once produced, EA, except for an insignificant fraction, reaches the small and then the large intestine undamaged, where, together with the EA produced by gut microbiota from ETs, is metabolized to UROs, which, in turn, are converted to their conjugates as schematized in Figure 4.

### 3.3. EA Chemical Reactivity

EA easily undergoes exothermic acid-base reaction and can be effortlessly sulfonated and nitrated by the corresponding acids [8].

As to the chemical reactivity, EA can undergo three general reaction types:(a)nitrosation (electrophilic aromatic substitution, non in vivo) of the electron rich aromatic rings, by reaction with sodium nitrite, mineral acid, and pyridine [29], to produce a red quinone oxime (λ max = 538 nm). This reaction is the basis for a spectrophotometric analysis of EA (Figure 5).(b)in vivo oxidation reactions i.e.,(i)EA oxidation by reactive FRs. Through this reaction, EA inhibits the dangerous effects of ROS and lipid peroxidation [29].(ii)EA oxidation by nucleophilic addition to the electrophilic epoxide named benzo[a]pyrene-7, 8-diol-9, 10-epoxide (BPDE) (Figure 6).

BPDE is a benzo[a]pyrene-derived carcinogen able to promote an electrophilic alkylation of genetic materials (DNA, RNA) with consequent genetic mutation of the cell. The reaction of EA with this electrophilic DNA-damaging agent has been proposed as possible mechanism for EA anti-carcinogenic effects [61]. (c)in vivo other reactions:(i)EA is able to interact with several important biological macromolecules such as DNA, exercising anti-mutagenic and anti-carcinogenic activity [62,63,64].(ii)EA can act as selective estrogen receptor modulator (SERM) with the possibility to work both as estrogenic and anti-estrogenic [65].(iii)as nuclear hormone receptor, working as a so called “Endocrine Disruptome” with antagonist or agonist activity [66].(iv)EA by interacting with polyphenol oxidase enzymes [29,67] can be oxidized to produce a 1,2-quinone able to develop acute cytotoxicity inside unhealthy cells, causing their death and ameliorating many not curable pathologies such as cancer [68].

## 4. The Common Sources of EA

EA is a common secondary metabolite in many medicinal plants and vegetables, where its free form is present at very low concentrations. Mainly, EA is present in glycoside forms, i.e., conjugated with a saccharide unit, such as glucose, rhamnose, arabinose, or in complex derivatives, as component of ETs. EA is a sub-fraction of ETs, in many fruits (pomegranates, persimmons, raspberries, black raspberries, wild strawberries, peaches, plums), in some nuts (walnuts, almonds), in seeds such as berry seeds, in vegetables [7,8], and in many species of medicinal plants, associated with health benefits and commonly ingested with a diet. An updated list of plants (43 species), where EA was isolated or only identified, is available in Table 7.

In the following Table (Table 8) fruits, vegetables, nuts, fresh or minimally processed food and beverages are listed, in which EA was found and frequently quantified.

Discrepancies are not unexpected as EA content values (even for the same plant material) can vary markedly depending on different extracting conditions [111,123,125], genetic diversity, climate, ripening stage soil, and storage conditions [126]. Kakadu (Australian indigenous Kakadu plum, *Terminalia ferdinandiana*) as well as other two Australian vegetables (i.e., Anise myrtle, *Syzygium anisatum* and Lemon myrtle, *Backhousia citriodora*) encompass a great amount of EA in the free form, thus representing an exception to the rule. Kakadu showed the highest free EA concentration [(228–14,020 mg/100 g of dried weight (DW)] [113,125,126,127].

## 5. EA Antioxidant Power: The Proposed Mechanisms of Action

A large variety of natural antioxidants are present in food and phenolic compounds encompass more than 8000 molecules, sharing the structural feature of presenting a phenol moiety.

EA hydroxyl groups and the lactone systems are home to hydrogen bonds, but can also act as electron acceptors and hydrogen donors. Consequently, EA is endowed with the capacity to accept electrons from different substrates and with the possibility to participate in antioxidant redox reactions, thus resulting a very efficient FRs scavenger [8].

Antioxidants can be considered chemical protectors classified on the basis of their mode of action [128]. Primary antioxidants (Type I, or chain breaking) are chemical species able to prevent oxidation by acting as free-radical scavengers, while secondary antioxidants (Type II, or preventive) act through indirect pathways and by retarding the oxidation process. In this case, antioxidants work through metal chelation, decomposition of hydroperoxides to non-radical species, by repairing of primary antioxidants with hydrogen or electron donation, by deactivating of singlet oxygen, impounding of triplet oxygen, and by absorbing UV radiation [128]. Since EA exerts its protective antioxidant effects through both primary and secondary ways of action, it can be classified as a multiple-function antioxidant.

Concerning the primary way of action, the reactions between EA and FRs are second order reactions and depend both on the concentration of EA and FRs, and on factors influencing their chemical structures such as the medium pH, polarity, and the reaction conditions. In general, the antioxidant capacity of EA is strongly influenced by the reaction medium and, in particular, it is reduced in solvents able to form hydrogen bonds with EA and it is improved in solvents favoring EA ionization to anion phenoxide [129]. In general, the antiradical properties of different natural and synthetic primary antioxidants with OH groups, derive from their capacity to donate hydrogen atom to a FR. Due to this transfer, non-radical species or a new radical, more stable and less reactive than the previous ones, form and possess the ability to exert antioxidant effects by several mechanisms.

The possible action mechanisms proposed for Type I antioxidants, including EA, include hydrogen atom transfer (HAT), proton coupled electron transfer (PCET), single electron transfer (SET), single electron transfer followed by proton transfer (SET-PT), sequential proton loss electron transfer (SPLET), radical adduct formation (RAF), and sequential proton loss hydrogen atom transfer (SPLHAT) [128,129,130,131,132,133]. Table 9 reports the chemical equations associated to these proposed mechanisms.

EA can exert antioxidant effects mainly through three of the above-mentioned reaction mechanisms.

The first is based on the SET reactions, while the second and third are based on the HAT and SPLHAT reactions, respectively. Although the result is the same, i.e., the inactivation of FRs to neutral, cationic, or anionic species, the kinetics and secondary reactions involved in the processes are different. The SET reactions involve the transmission from EA of single electrons with formation of a more stable EA^•^ and charged species. While, the HAT and SPLHAT reactions involve the inactivation of FRs through the donation of a hydrogen atom with formation of EA^•^ and uncharged species. In this regard, EA can react with RONS through HAT reaction, breaking the chain leading to the generation of new reactive toxic species, thanks to its different hydroxyl groups, which are good hydrogen donors [134,135,136,137,138]. In particular, the reaction mechanism of EA with ROO^•^ consists of a transfer of the hydrogen cation from the EA hydroxyls to the radical species, forming a transition state of an H-O bond with one electron. On the other hand, the hydroxyl groups can interact with the π-electrons of the benzene ring providing molecules endowed with the ability to generate free long-living radicals stabilized by delocalization, able to interfere and modify radical-mediated oxidation processes, by SET reactions.

In vitro tests have shown that the type and polarity of solvents strongly influence the antioxidant capacity and the mechanism of action of EA.

The alcohols may act either as acceptors of hydrogen bonds, thus reducing EA antiradical effects by hydrogen atom transfer (HAT) reactions, or favoring the ionization of the EA to anion phenoxides, which can react rapidly with the peroxyl radicals, through an electron transfer, thus improving EA radical scavenging activity by SET reactions [129].

Since EA is also a Type II antioxidant, it exerts its effects against FRs thanks to its ability in inhibiting the endogenous production of oxidants and in particular of radical hydroxyl (^•^OH), which is the most reactive and electrophilic of the oxygen-based radicals.

^•^OH is the factor main responsible for tissue and DNA damage and therefore, its inhibition is of primary importance for reducing OS generated from the metal-catalyzed Fenton reaction and the HWR (Table 3), according to Equations (1)–(4), respectively, involving the reduced forms of Fe and Cu.
Fe(II) + H_2_O_2_ → Fe(III) + OH^−^ + ^•^OH(1)Cu(I) + H_2_O_2_ → Cu(II) + H^−^ + ^•^OH(2)Fe(III) + O_2_^•^^−^ → Fe(II) + O_2_(3)Cu(II) + O_2_^•^^−^ → Cu(I) + O_2_ (Fenton)(4)

EA together with melatonin and its metabolites, *N*^1^-acetyl-*N*^2^-formyl-5-methoxykynuramine (AFMK), cyclic 3-hydroxymelatonin (3OHM), quercetin, and luteolin are exemplary antioxidants able to exert their protection by chelating and subtracting metal as Fe^2+^, Fe^3+^, and copper ions involved in the production of FRs, thus preventing the oxidation of LDL [128,139,140].

Phenolic structures as that of EA, for the presence of the hydrophobic benzenoid rings and for the capability of the phenolic hydroxyl groups to form hydrogen-bonding interactions, could also interact with enzymes involved in radical generation, such as various cytochrome P450 isoforms, lipoxygenases, cyclooxygenase, and xanthine oxidase, thus inhibiting RONS over production [134]. Additionally, EA can work synergistically with other endogenous and exogenous antioxidants, such as ascorbic acid, β-carotene, and β-tocopherol, thus increasing their effectiveness and can regulate intracellular glutathione levels [134].

Regardless, it is necessary to remember that due to some of their hydroxyl groups, phenolic antioxidants including EA, under certain conditions, including high dosage, high concentrations of transition metal ions, alkali pH, and the presence of oxygen molecules, can act as pro-oxidants [141].

These groups may sometimes induce significant DNA damage in the presence of Cu (II) or may create ROS through the reduction of Cu (II) → Cu. The pro-oxidant activity, peculiar of small polyphenols, can trigger apoptosis in cancer cells [142]. In contrast, large molecular-weight phenols, such as ETs, have little or no pro-oxidant properties [143].

## 6. The Urolithins (UROs) System

### 6.1. Structure and Chemistry

The term urolithins (UROs) refers to a very large family of metabolites produced from free EA not absorbed in the stomach and to a lesser extent by the mammalian gut-microbiota. UROs are dibenzopyran-6-one derivatives with different hydroxyl substitutions, which can be considered a combination of benzocoumarin and iso-benzocoumarin, including mainly urolithin A (URO A), urolithin B (URO B), and their isomeric forms Iso URO A and Iso URO B (Figure 4, Section 3.2).

The chemical and physical computed properties of URO A and URO B are reported in Table 10.

### 6.2. Formation Pathway and Metabolism of UROs

From a chemical point of view, UROs form through the opening and decarboxylation of one of EA lactone rings and the sequential and progressive removal of hydroxyls from different positions (dehydroxylase activities). The gut microbiota metabolic activity is responsible for these in vivo transformations. In humans, the complete metabolism of EA leads to the production of two UROs subtype, i.e., URO A and B and of their structural isomers Iso URO A and Iso URO B (Figure 4, Section 3.2).

In particular, after the opening of one of the EA lactone rings, catalyzed by a lactonase to give Luteic Acid (LA), a first decarboxylation occurs thanks to a decarboxylase which provides the first metabolite urolithin M-5 (pentahydroxy-urolithin). From this, in the small intestine by removal of one hydroxyl group from different positions tetrahydroxy-urolithin isomers (URO D, URO M-6) are produced, subsequently trihydroxy-urolithins (URO C, URO M-7) are synthetized thanks to the removal of a second hydroxyl and finally dihydrox-urolithins (URO A and Iso URO A) are obtained after the removal of a third one. In addition, monohydroxy-urolithins (URO B and Iso URO B) are also detected in the large intestine, particularly in those cases in which Iso URO A is produced. Further degradation of UROs to remove the second lactone ring has not been reported so far [43] although it should not be discarded [38].

Thanks to their chemical structure, UROs can be easily absorbed, circulate in plasma, reach tissues, including the central nervous system [9], exert their activities and then, they are incorporated into enterohepatic circulation or are conjugated with methyl, glucuronic acid, or sulfate and are eliminated with feces and urine. In this regard, results obtained in animals treated with ETs-rich diet (food and beverages) lead to the identification and quantification of the metabolites in plasma, urine, feces, and that enter tissues. The prevalent metabolites detected in plasma and urine correspond to URO-A, Iso URO-A, and URO-B, mainly conjugated with glucuronic acid [43]. The available data regarding UROs concentrations in plasma, urine, and different tissues are sometimes very different and affected by an overestimation, depending on the analytical methods adopted. Regardless, the glucuronide conjugated of URO A, B, and Iso URO A has been detected in plasma in the range of 0.045–35 µM and in the urine up to 100 µM, even if concentrations of 5330 and 6185 µM have been also reported for URO A and URO B, respectively.

Studies for evaluating UROs concentration in the tissues were mainly performed on rats and pigs and showed a strong dependence on the tissue target and a very large range of 100–1050 ng/g with decreasing accumulation from prostate, intestine, liver, kidney, to lung [43].

### 6.3. Influence of Individual Metabotype on UROs Production

As described for isoflavones, whose metabolism varies among individuals which may be either *O*-desmethylangolensin (ODMA) ‘producers’ or ‘non-producers’ [46], the EA metabolism and consequently the type and quantity of UROs produced depends on the gut microbiota composition that in turn depends on the individual state of health and age, on the environmental and life conditions, and on human metabotype (UM) [44,47].

Three different UMs exist: UM 0 (no UROs producers), UM A (producers of URO A), and UM B (producers of URO B and/or Iso URO A). UM dictates gut microbiota composition and, therefore, UROs production. In addition, due to the influence of age, health status, and obesity, UM can change also across the lifespan, consequently UROs type production and the amount of UROs produced for a single individual can change across the lifespan and with age [9,47]. In this regard, in vitro studies concerning metabolism of EA by *Gordonibacter urolithinfaciens* and *G. pamelaea* (*Gordonibacter* genera belonging to *Coriobacteriaceae* family) showed the production of tri-hydroxyl derivative URO C only [45]. Monitoring this metabolism by HPLC-MS, it was observed that such bacteria are able to produce only species of UROs not completely transformed to the final metabolites URO-A, Iso URO-A, and URO-B. In particular, the analysis showed the sequential production of URO-M5, URO-M6 up to URO-C from which, also after longer incubation periods, no further hydroxyl was removed by these microbes [43]. These findings demonstrated that the type of UROs produced depends on the type of bacteria which form the gut microbiota and that for the complete transformation of EA into URO-A and URO-B, other bacteria, still unknown, are necessary.

Regardless, it is also possible that *Gordonibacter* species manage the complete catabolism of EA to final UROs only at physiological conditions found in vivo, which might be critical for their functioning. Differently, other genera from *Coriobacteriaceae* family tested by Selma et al. in 2014, were not able to produce any kind of UROs [45].

In a study on healthy human volunteers subjected to an acute consumption of 800 mg of pomegranate extract, it was reported that URO A and B were detected after 8 and 24 h in, altogether, three of the 11 subjects, whereas URO A-glucuronide was detected in six of the 11 subjects. Hydroxyl-URO A was found in three subjects at several time points from 2 to 24 h and URO A-glucuronide was found over a period of 2–24 h in six of the subjects, stating that UROs type production varies from individual to individual and that they can circulate in plasma up to 24 h after the intake [144].

The dependence of EA metabolism and types of UROs produced by the human gut microbiota composition and UM was confirmed also by studies on ex vivo cultures [145].

In order to establish the impact of aging on the distribution of UMs and the potential correlation with obesity, lifestyle, and health status, a study was performed on a large Caucasian cohort (*n* = 839), aged from 5 to 90 years. The findings from this study confirmed, for the first time, that aging is the main factor, followed by health status and weight, that determinates the gut microbiota composition and the type of UROs produced, with potential consequences for human health [47].

### 6.4. Not All Living Species Produce UROs

The metabolism, which transforms EA in UROs, does not occur in all living species, due to the incapability to perform such metabolic pathway characterizing the microbiota of some species. Regardless, the UROs production from ETs has been reported for different animals (Table 11).

Concerning UROs production in rats, ETs from strawberries with different degrees of polymerization, showed different metabolisms and, in particular, ETs with low degree of polymerization were metabolized to URO A and considerable amounts of nasutins, while ETs with high degrees of polymerization were converted, probably by the action of *Eubacterium* only to nausitins [43,147].

Regardless, UROs are metabolites present in humans [38,44,148] pigs [38], in beavers, mice, sheep, and cows [146]. Additionally, in castoreum and in pigs fed oak acorns the removal of EA phenolic hydroxyls without opening the lactone ring gives rise to a relevant number of nasutin metabolites [146].

### 6.5. Mechanisms of Antioxidant Effects of UROs

Among UROs, URO A, Iso URO A, and URO B are the most common UROs found in humans and animals. Differently from EA, that is both a type I and a type II antioxidant able to exert antioxidant effects both by SET reactions and HAT ones, UROs have been identified as potent antioxidants only by the oxygen radical absorbance capacity (ORAC) chemical in vitro assay [149]. ORAC measures antioxidant inhibition of oxidation induced by peroxyl radicals and therefore reflects classical radical chain-breaking antioxidant activity by HAT mechanism [150].

In other assays based on the SET mechanism, such as 2,2-diphenysl-1-picrylhydrazyl (DPPH), ferric reducing ability of plasma (FRAP) and 2,2’-azino*bis*-(3-ethylbenzothiazoline-6-sulfonic acid) (ABTS^•^^+^) assays, UROs proved to possess questionable antioxidant activity. In this regard, URO A, tested as direct radical scavenger (type I antioxidant), showed an IC_50_ of 152.66 µM in the DPPH test [150], i.e., 23-fold less active than EA which proved an IC_50_ of 6.6 µM [151].

In effect, unconjugated UROs do not have other functional groups in addition to two (URO A) or even one (URO B) phenolic hydroxyl groups, that preferentially act as hydrogen donors, favoring HAT reactions and the electron-withdrawing carboxyl group, being part of a lactone ring, cannot promote the SET mechanism. All this evidence suggests that the antioxidant activity of UROs is mediated exclusively by the HAT mechanisms [149].

Although UROs play a minimal role as a direct radical scavenger, their cytoprotective action has to be attributed mainly to the improvement of the cellular antioxidant defenses and to their activity as oxidases inhibitors [150].

## 7. EA or UROs: An Open Debate

The question about who really is responsible between EA and UROs for the health benefits deriving from the intake of ETs-rich foods, has not been clarified at all, and it is still much debated and under investigation so far. Regardless, the hypothesis that UROs are the actual bioactive compounds, firstly launched by Cerda et al. in 2005 [152], it is now the most consolidated and trustworthy one. This assumption finds justifications in the fact that practically only UROs and their phase II conjugates, are adsorbed and are able to circulate in the blood and to reach the different target tissues, where they may interact with the cell machinery and trigger different molecular and cell responses. Differently, the rather non-sense blood concentrations achieved by EA are not enough for justifying benefits coming from ETs-rich plants intake.

On the other hand, the knowledge of the distribution of UROs in human tissues is still too limited and incomplete to elect UROs as promising molecules for clinical applications devoted to prevent and/or treat OS and ageing-related disorders, due to the complexity of performing such investigations [38]. The current knowledge concerning the in vivo concentrations of UROs and their conjugates induce to reflect whether the in vitro reported UROs biological effects may be of real relevance in vivo [43].

In order to clarify which fluids and/or biological tissues are that ones where UROs, their conjugates or nusatins can be found after the intake of ETs rich foods, ETs, or EA, a complete study was developed, whose results are listed in Table 12 [43]. In the table the acronym URO was left out.

### Hazardous Implications Related to UROs Exposure

Results obtained from a randomized clinical trial showed that the inter-individual variability in the improvement of cardiovascular risk biomarkers, in overweight-obese individuals consuming pomegranate, depends on different UMs and type of UROs produced.

In particular, a high cardiovascular risk was associated to the UMB that dictates for a gut microbiota composition that determinates a higher production URO B and Iso URO A, rather than URO A or no production [47]. In fact, in this regard, the positive effects of UROs are obscured by a negative side effect Iso URO A and URO B [47].

In recent years, the dual redox behavior of natural polyphenols has increasingly captured the interest of researchers and the need to investigate their pro-oxidant capabilities in addition to the antioxidant effects has become urgent [149]. In this regard, UROs have received contradictory reports on their antioxidant capacity, and their pro-oxidant properties have been recently studied.

The redox properties of URO A and URO B, have been investigated by using more than one assay method including ORAC assay, three cell-based assays, copper-initiated pro-oxidant activity (CIPA) assay, and cyclic voltammetry, and the findings unveiled that UROs, although strong antioxidants in the ORAC assay, are mostly pro-oxidants in cell-based assays [149].

Citing Rahal et al., 2014 [153]: “Pro-oxidant refers to any endobiotic or xenobiotic that induces oxidative stress either by generation of ROS or by inhibiting antioxidant systems”. As reported, the pro-oxidant activity of natural polyphenols depends on their dosage and on the presence of proper amounts of metal ions [154], but while small polyphenols can exhibit considerable pro-oxidant activity, large molecular-weight phenols, such as ETs, have little or no pro-oxidant properties [144].

In this context, the tendency of small polyphenols to exert pro-oxidants effects is considerable in the presence of high concentrations of transition metal ions, such as Cu^2+^ or Fe^2+^. A proposed mechanism suggests that, firstly, UROs reduce Cu^2+^ to Cu^+^ and, subsequently, Cu^+^ is re-oxidized in a Fenton-like reaction by the action of H_2_O_2_ or O_2_, leading to the production of oxygen radicals [149]. Considering that in living cells, a small amount of hydrogen peroxide is produced as a result of cellular metabolism, and that transition metal ions are available, the proposed scenario is without a doubt feasible.

## 8. Authors Opinions, Future Perspectives and Conclusions

Hippocrates, over 2500 years ago, coined the phrase “*Let food be the medicine and medicine be the food*” and in the past, folk medicine, that made use of spices, plants, herbs, fruits, and seeds for the treatment of several diseases, gave it a considerable applicative importance.

Some foods are more than fuel for the organism, and could contain nutrients as ETs and EA, that proved to possess activities essential to preclude diseases and to own high potentials for being effectively used as a sort of medicine to prevent and/or enhance illnesses.

Nowadays, an extensive literature, mainly based on in vitro tests, reports the several EA health properties, based on its multi-target antioxidant effects, able to counteract OS and aging-related diseases. As a consequence, a worldwide market concerning the production of plants concentrated extracts, EA-enriched foods, EA-based functional foods or food supplements, rapidly has been developed, with the claim to provide tools to increase the daily intake of EA and with a global conspicuous involvement of large capitals. In this regard, the actual potential, effectiveness, and safety of EA-enriched products should be discussed and a more wide-ranging debate is needed.

Since EA is practically not soluble and only an insignificant fraction of the EA contained in these products will actually be absorbed in GIT, essentially, all EA ingested will be metabolized by gut microbiota to UROs, which, easily absorbed, will circulate in blood and reach cells and tissues. Ironically, by the intake of EA-enriched products, in place of improving EA in vivo concentration that of UROs is increased, both in plasma and in tissues.

Nevertheless, depending on the age of individuals, their health conditions, and the composition of their gut microbiota, the types of UROs produced and their concentration can vary and the individual variability in the responses to UROs exposure is unpredictable and may lead to heterogeneous comebacks that could promote also adverse effects, such as cardiovascular disorders. Furthermore, by exerting both antioxidant and pro-oxidant activities, high concentration of UROs may cause DNA damage and apoptosis, leading to the development of OS-related pathologies such as cancer. Consequently, EA-enriched products’ commercialization should be subject to more careful control and regulation, and after their intake, URO concentrations and tissues distributions, should be monitored and kept under continuous control, individual by individual.

An uncontrolled large production and distribution of EA-enriched foods and food supplements represent an outstanding problem, which could have heavy negative repercussions on human health rather than positive.

Regardless, further detailed and scrupulous toxicity evaluations are necessary both for EA and UROs, before their therapeutic application in humans could be possible. Clinical investigations need to verify the interesting preclinical findings and to validate if the very good results observed in animals are confirmed in humans.

It is relevant to understand also the inter-individual variability of the population in response to EA and ETs-containing foods, for a better organization of a healthy diet and for establishing the most suitable dosages of polyphenolic nutraceuticals and foods rich in ETs and EA for each person.

As far as the authors’ opinion is concerned, it seems anyway conceivable to propose that EA remains the robust compound worthy of further extensive investigations.

In relation to its chemical structure and antioxidant mechanisms, EA might be either a safe nutraceutical, an innovative therapeutic, or a template molecule for the development of novel drugs able to fight RONS for promoting human health.

Finally, although attractive, the hypothesis that declares UROs as the actual substances responsible of beneficial effects coming from the ingestion of EA-rich foods is far from being clearly demonstrated, because studies investigating the in vivo EA effects are limited or even missing.

There is no evidence for attributing with certainty also to EA an in vivo activity, but it is equally incorrect to affirm the opposite and attribute in vivo activity to UROs only.

## Figures and Tables

**Figure 1 antioxidants-09-00707-f001:**
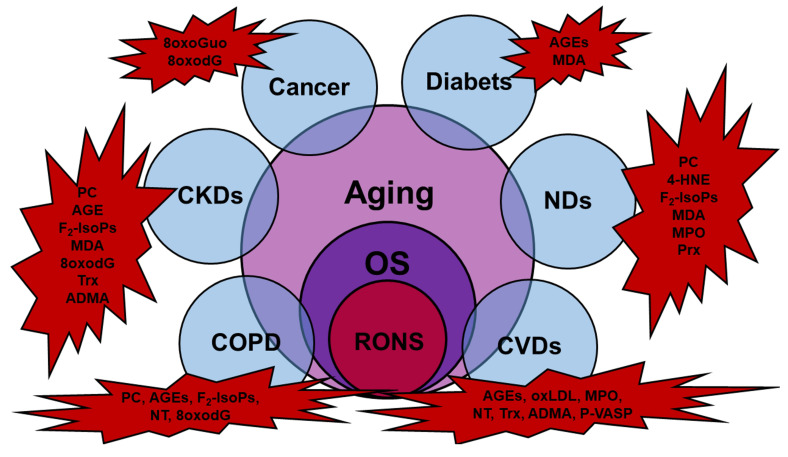
The most reported diseases interconnected to oxidative stress (OS) and to OS-related aging, with evidence of the relative species detected and considered OS biomarkers (red explosions). Abbreviations: 4-HNE, trans-4-hydroxy-2-nonenal; 8oxodG, 7,8-dihydro-8-oxo-2′-deoxyguanosine; 8oxoGuo, 7,8-dihydro-8-oxoguanosine; ADMA, asymmetric dimethyl l-arginine; AGEs, advanced glycation end products; F_2_-IsoPs, F_2_-isoprostanes; MDA, malondialdehyde; MPO, myeloperoxidase; NT, nitrotyrosine; oxLDL, oxidized low-density lipoprotein; PC, protein carbonyl; Prx, peroxiredoxins; P-VASP, phosphorylated vasodilator-stimulated phosphoprotein; Trx, thioredoxin.

**Figure 2 antioxidants-09-00707-f002:**
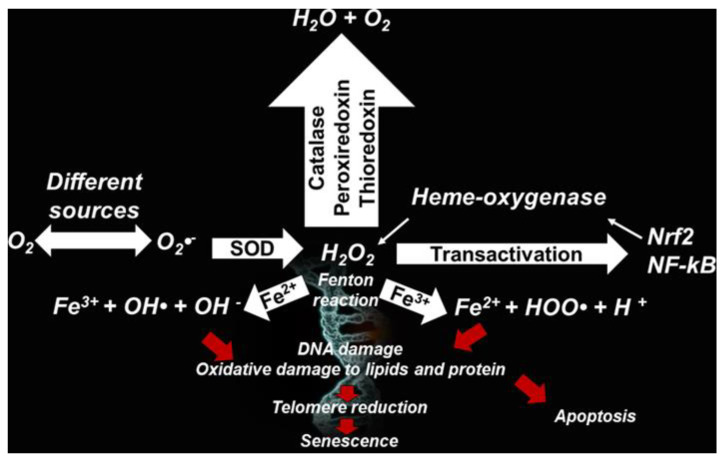
Pathways of reactive oxygen species (ROS) production.

**Figure 3 antioxidants-09-00707-f003:**
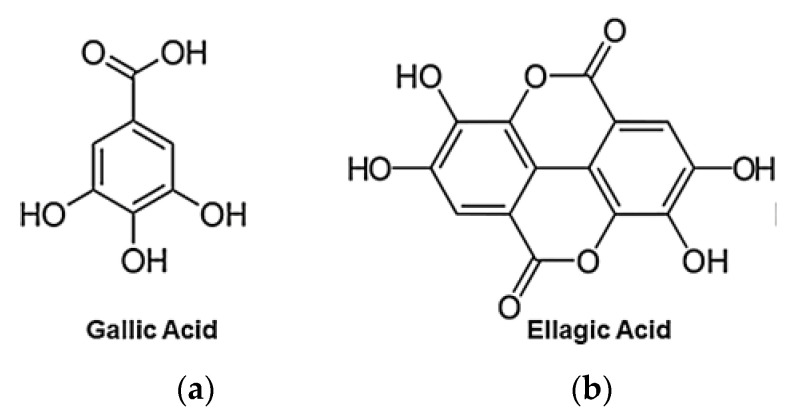
Structure of two metabolites of ETs: (**a**) gallic acid (GA); (**b**) ellagic acid (EA).

**Figure 4 antioxidants-09-00707-f004:**
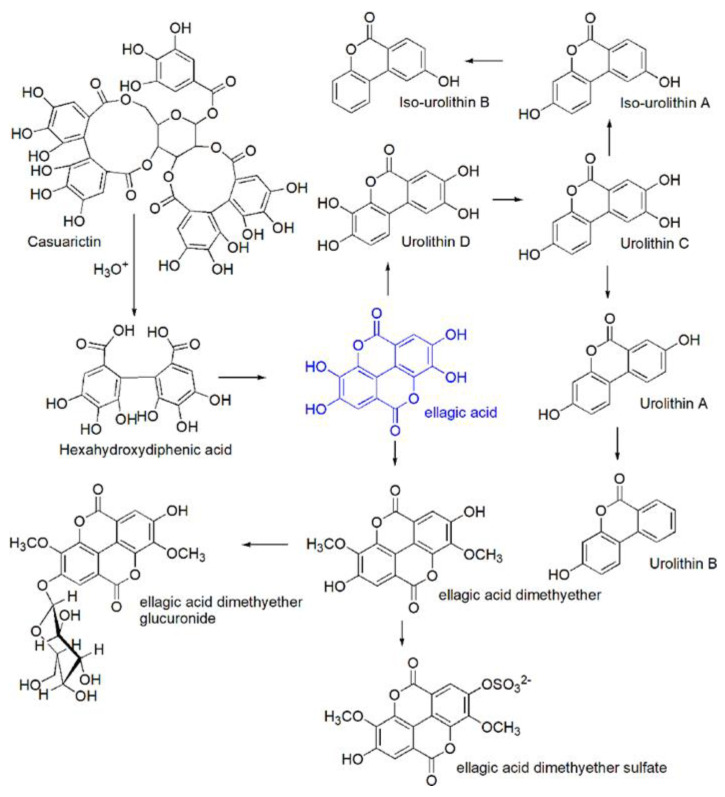
In vivo EA formation and metabolism.

**Figure 5 antioxidants-09-00707-f005:**
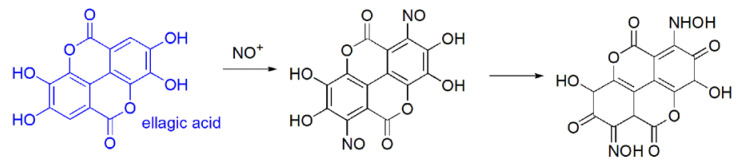
A hypothesized EA non in vivo nitrosation.

**Figure 6 antioxidants-09-00707-f006:**
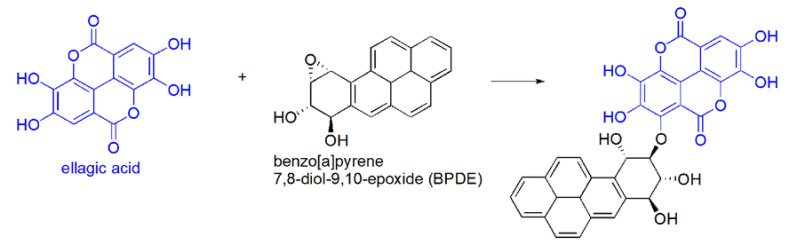
A hypothesized EA in vivo reaction.

**Table 1 antioxidants-09-00707-t001:** Components influenced by reactive oxygen and nitrogen species (RONS) and effects provoked [50]. ↑ Means increase; ↓ means decrease/reduction.

Components Influenced by RONS	Effects
Regulation of mTORC1	Impairments in cell growth and metabolism
Production of IL-1α	Pro-inflammatory state ↑NFκB ↑Epithelial–mesenchymal transition ↑Tumor metastatic progression
Induction of MMPs expression	Cancer, Alzheimer’s, atherosclerosis, osteoarthritis, lung emphysema
↓FOXO proteins activity	↓IGF-1-mediated protection from OS
↓Sarco/endoplasmic reticulum Ca^2+^-ATPase activity	Cardiac senescence
↓Sirtuins activity	↑RONS by SOD inhibition Pro-inflammatory state, tumorigenic effect ↑TNFα and NFκB ↑Proto-oncogene c-Jun and c-Myc
Regulation of p16INK4a/pRB and proteins p53/p21	Senescence

Abbreviations: mTORC1, mammalian target of rapamycin complex 1; FOXO, forkhead box O proteins; IGF-1, insulin-like growth factor 1; SOD, superoxide dismutase; TNFα, tumor necrosis factor alpha; p16INK4a, cyclin-dependent kinase inhibitor; pRB, retinoblastoma protein.

**Table 2 antioxidants-09-00707-t002:** Endogenous and exogenous sources of RONS and possible ROS and reactive nitrogen species (RNS).

Endogenous Sources		Exogenous Sources	Reactive Species
Enzymatic	Non-Enzymatic		
NOX	Respiratory chain	Air	O_2_^•^^−^ H_2_O_2_ ^•^OH O^•^OH ONOO^•^ NO_2_^•^ CO_3_^•^^−^ NO^•^ ONOOCO_2_^−^ NO^2+^ ONOOH N_2_O_3_ ONOO^−^ ONOOCO_2_^−^
MPO	Glucose auto-oxidation	Water pollution	
Lipoxygenase	NAD^•^	Tobacco	
Angiotensin II	Semiquinone radicals	Alcohol	
NOS	Radical pyridinium	Heavy/transition metals	
Xanthene oxidase	Hemoproteins	Drugs	
Cyclooxygenase		Industrial solvents	
FpH^•^		Cooking	
		Radiation	

Abbreviations: NADPH, nicotinamide adenine dinucleotide phosphate; MPO, myeloperoxidase; NOX, NADPH oxidase; NAD, nicotinammide adenina dinucleotide; Fp, flavoprotein enzymes.

**Table 3 antioxidants-09-00707-t003:** The most representative radicals and the reactive specie ONOOCO_2_^−^ produced in biological aerobic systems with sources and functions.

Radical	Source	Function
O_2_^•^^−^	Enzymatic process Autoxidation reactions Non-enzymatic electron transfer reactions	Reducing agent of iron complexes such as cytochrome-c Oxidizing agent to oxidize ascorbic acid and α-tocopherol
HO_2_^•^	Protonation of O_2_^•^^−^	HOO^•^ initiates fatty acid peroxidation
HO^•^	H_2_O_2_ generates HO^•^ through the metal-catalyzed Fenton reaction and the Haber Weiss recombination (HWR)	HO^•^ reacts with both organic and inorganic molecules (DNA, proteins, lipids, carbohydrates)
NO^•^	*L*-arginine (substrate) NADPH (electron source) Nitric oxide-synthase	Intracellular second messenger Stimulates guanylate cyclase and protein kinases Causes smooth muscle relaxation in blood vessels
NO_2_^•^	Protonation of ONOO^−^ Homolytic fragmentation of ONOOCO_2_^−^	Acts on the antioxidant mechanism Decreases ascorbate and α-tocopherol in plasma
ONOO^•^	Reaction of O_2_ with NO^•^	Oxidizes and nitrates methionine and *L*-tyrosine residues in proteins Oxidizes DNA to form nitroguanine
CO_3_^•^^−^	(SOD)-Cu^2+^-^•^OH reacts with HCO_3_^-^ to generate CO_3_^•^^−^	Oxidizes proteins and nucleic acids
**Reactive specie**	**Source**	**Function**
ONOOCO_2_^−^	Reaction of ONOO^−^ with CO_2_	Promotes nitration of tyrosine of the oxyhemoglobin via free radicals

**Table 4 antioxidants-09-00707-t004:** Oxidative modification of cellular macromolecules: reactions involved and markers of OS produced in such way.

Cellular Macromolecules	Reactions	OS Biomarkers	Ref
Proteins	RNS with free or within polypeptide sequences *L*-tyrosine	Nitrotyrosine (NT)	[53]
	Fenton reaction of oxidants with *L*-lysine, *L*-arginine, *L*-proline, *L*-threonine	PC	[54]
Proteins and lipids	Michael-addition of aldehydic lipid oxidation products to *L*-lysine, *L*-cysteine, *L*-histidine	PC	[54]
Proteins and lipids	Complex oxidative process	oxLDL	[55]
Proteins and carbohydrates	Glyco-oxidation between *L*-lysine amino groups and *L*-arginine carbonyl groups linked to carbohydrates	AGEs (*N*-ε-carboxymethyl-lysine pentosidine glucosepane)	[56]
Lipids	Hydroxyl and peroxyl radicals-mediated lipid peroxidation of poly-unsaturated fatty acids (linoleic, arachidonic acids)	4-HNE MDA F_2_-IsoPs	[53]
DNA	Mutagenic oxidation	2-hydroxy adenine 8-oxoadenine 5-hydroxycytosine Cytosine glycol Thymine Glycol 8-oxoGuo 8-oxodG	[57]

**Table 5 antioxidants-09-00707-t005:** Endogenous and exogenous molecules for counteracting free radicals (FRs) and reactive species toxicity. ↑ Means increase; ↓ means decrease/reduction.

Endogenous Molecules		Actions	Exogenous Molecules	Effect
Not enzymatic	Vitamin E Vitamin C carotenes Ferritin Ceruloplasmin selenium GSH manganese Ubiquinone zinc Flavonoids coenzyme Q Melatonin Bilirubin taurine cysteine Albumin Uric acid	Interact with RONS and terminate the free radical chain reactions	Vitamin C	↓O_2_^•^ ↓^•^OH
			Vitamin E	↓Lipid peroxidation
			Resveratrol Phenolic acids Flavonoids	↓O_2_^•^ ↓^•^OH ↓Lipid peroxidation
			Oil lecithin	↓O_2_^•^ ↓^•^OH ↓Lipid peroxidation
			Selenium Zinc	Antioxidant
			Acetylcysteine	Antioxidant
Enzymatic	SOD	Converts O_2_ to H_2_O_2_ ↓hydroxyl radical production		
	Catalase (CAT)	Decomposes H_2_O_2_ to H_2_O+O^2^ ↓hydroxyl radical production		
	GSH-Px	Converts peroxides and hydroxyl radicals into nontoxic forms by the oxidation of reduced glutathione (GSH) into glutathione disulphide		
	GR	Converts glutathione disulphide to GSH		
	GSTs	Catalyze the conjugation of GSH to xenobiotic substrate detoxification		
	G6PD	Catalyzes the dehydrogenation of glucose-6-phosphate to 6-phosphoglucono-Δ-lactone		
	Nrf2	Regulates the expression of antioxidant proteins		
	ARE	Encodes for detoxification enzymes and cytoprotective proteins		
	NQO1	Catalyzes the two-electron reduction of quinones and quinonoid compounds to hydroquinones		
	MSR	Carries out the enzymatic reduction of the oxidized form of methionine to methionine		

Abbreviations: GSH, reduced glutathione; ARE, antioxidant response element; NQO1, NAD(P)H quinone acceptor oxidoreductase (DT-diaphorase); GSTs, glutathione S-transferases; G6PD, glucosio-6-fosfato dehydrogenase.

**Table 6 antioxidants-09-00707-t006:** Chemical and physical properties of EA [40,58].

Physicochemical Identifiers	Descriptive Data
Chemical Name ^1^	Ellagic Acid
CAS number	476-66-4
Molecular formula	C_14_H_6_O_8_
Molecular weight	302.194 g/mol
Hydrogen bond donor count	4
Hydrogen bond acceptor count	8
Covalently bonded unit count	1
Form/color	Cream colored needles from pyridine Yellow powder
Melting point	>360 °C
Density	1.667 at 18 °C
Dissociation constants	pKa_1_ = 6.69 (phenol) pKa_2_ = 7.45 (phenol) pKa_3_ = 9.61 (phenol) pKa_4_ = 11.50 (phenol)
Solubility	Slightly soluble in alcohol and water Insoluble in ether Soluble in alkalis and pyridine
Vapor pressure	2.81×10^−15^ mm Hg at 25 °C
Spectral properties	UV max (ethanol): 366, 255 nm

^1^ traditional IUPAC name.

**Table 7 antioxidants-09-00707-t007:** Presence of ellagic acid in several medicinal plants.

Plant Species	Family	Ref.
*Acalypha hispida* Burm.f.	Euphorbiaceae	[69]
*Acca sellowiana* (O.Berg) Burret ^1^	Myrtaceae	[70]
*Baccharis inamoena* Gardner ^1^	Compositae	[71]
*Camellia nitidissima* C.W.Chi ^1^	Theaceae	[72]
*Campomanesia adamantium* (Cambess.) O.Berg	Myrtaceae	[73]
*Canarium album* (Lour.) DC. ^1^	Burseraceae	[74]
*Carpobrotus edulis* (L.) N. E.Br.	Aizoaceae	[75]
*Castanea crenata* Sieb. and Zucc.	Fagaceae	[76]
*Clematis ispahanica* Boiss.	Ranunculaceae	[77]
*Clematis orientalis* L.	Ranunculaceae	[77]
*Clerodendrum infortunatum* L. ^1^	Lamiaceae	[78]
*Cornus officinalis* Siebold and Zucc.	Cornaceae	[79]
*Elaeagnus rhamnoides* (L.) A.Nelson ^1^	Elaeagnaceae	[80]
*Euterpe edulis* Mart.	Arecaceae	[81]
*Eugenia uniflora* L.	Myrtaceae	[82]
*Euphorbia pekinensi**s* Rupr.	Euphorbiaceae	[83]
*Geum urbanum* L.	Rosaceae	[84]
*Gymnanthes lucida* Sw. ^1^	Euphorbiaceae	[85]
*Juglans regia* L.	Juglandaceae	[86]
*Lafoensia pacari* A. St.-Hil.	Lythraceae	[87]
*Myrciaria floribunda* (H.West exWilld.) O.Berg	Myrtaceae	[88]
*Myrtus communis* L.	Myrtaceae	[89]
*Nephelium lappaceum* L.	Sapindaceae	[90]
*Pandiaka angustifolia* (Vahl) Hepper	Amaranthaceae	[91]
*Phyllanthus acuminatu**s* Vahl	Phyllanthaceae	[92]
*Pleurotus eryngii* (DC. ex Fr.) Quel	Pleurotaceae	[93]
*Plinia cauliflora* (Mart.) Kausel ^1^	Myrtaceae	[94]
*Plinia coronata* (Mattos) Mattos ^1^	Myrtaceae	[95]
*Plinia peruviana* (Poir.) Govaerts	Myrtaceae	[96]
*Potentilla anserina* L.	Rosaceae	[97]
*Psidium brownianum*	Myrtaceae	[98]
*Quassia undulata* (Guill. and Perr.) D.Dietr.	Simaroubaceae	[99]
*Tetrapleura tetraptera* (Schum. and Thonn.) Taub.	Leguminosae	[99]
*Salacia chinensis* L.	Celastraceae	[100]
*Sambucus lanceolata* R.Br.	Adoxaceae	[101]
*Sanguisorba officinalis* L.	Rosaceae	[102]
*Sedum roseum* (L.) Scop. ^1^	Crassulaceae	[103]
*Sterculia striata**A.* St.-Hil. and Naudin	Malvaceae	[104]
*Syzygium calophyllifolium* (Wight) Walp.	Myrtaceae	[105]
*Syzygium cumini* (L.) Skeels	Myrtaceae	[106]
*Terminalia chebula* Retz.	Combretaceae	[107]

^1^ These plants are cited with the present name according to “The plant list. A working list of all plant species”. Available at http://www.theplantlist.org/.

**Table 8 antioxidants-09-00707-t008:** Contents of ellagitannins (EA) in fruits, nuts (mg/100 g of fresh weight), in beverages (mg/L), and in seeds (mg/g seed). The total EA content is estimated after acid hydrolysis so as to include ETs in the estimation [33].

Food/Fruits	EA Content	Ref.
Red raspberry (Ottawa)	70.8 ± 2.8 mg/100 g	[108]
Blackberries	150.0 ± 12.0 mg/100 g	[109]
Cranberry	12.0 ± 0.4 mg/100 g	[109]
Blackberries	150.0 ± 12.0 mg/100 g	[109]
Arctic bramble	390 mg/100 g	[110]
Raspberry (wild)	270 mg/100 g	[110]
Yellow raspberry	1900 mg/100 g	[110]
Black raspberries	90 mg/100 g	[111]
Boysenberries	70 mg/100 g	[111]
Evergreen blackberries	60 mg/100 g	[111]
Marionberries	73 mg/100 g	[111]
Cloudberry	1090–1423 mg/100 g	[108]
	360 mg/100 g	[110]
	315.1 mg/100 g	[112]
Raspberry	1692–1794 mg/100 g	[108]
	150.0 ± 10.0 mg/100 g	[109]
	263.7 mg/100 g	[112]
Rose hip	109.6 mg/100 g	[112]
Sea buckthorn	1.0 mg/100 g	[112]
Strawberry, Honeoye	77.6 mg/100 g	[112]
Strawberry, Polka	68.3 mg/100 g	[112]
Kakadu		
*Puree–2014 (month 0)*	1496 ± 76 mg/100 g	[113]
*Puree–2014 (month 3)*	1165 ± 24 mg/100 g	[113]
*Whole fruit–2014 (month 0)*	1726 ± 334 mg/100 g	[113]
*Whole fruit–2014 (month 3)*	1214 ± 192 mg/100 g	[113]
*Average whole fruit (indiv.)*	976 ± 223 mg/100 g	[113]
*Average leaves (indiv.)*	5848 ± 1046 mg/100 g	[113]
Pomegranate seeds pulp	980–2960 mg/100 g	[114]
Pomegranate dry peels	8300–26,300 mg/100 g (pressurized water extraction)	[115]
	137–6310 mg/100 g (different solvents)	[116]
Pomegranate whole fruit	26–2497 mg/100 g (different solvents)	[116]
Pomegranate marcs ^1^	216–352 mg/100 g (different methods)	[117]
Red raspberry	103.0 ± 3.3 mg/100 g	[109]
	47 mg/100 g	[111]
	160 mg/100 g	[110]
Strawberries	81–184 mg/100 g	[108]
	63.0 ± 9.0 mg/100 g	[109]
	65–85 mg/100 g	[110]
	31 mg/100 g	[118]
Strawberry, Jonsok	79.9 mg/100 g	[112]
	40.3 ± 7.5 mg/100 g	[119]
Strawberry jam	24.5 mg/100 g	[112]
	17–29.5 mg/100 g	[120]
Nuts		
Pecan	330.3 mg/100 g	[109]
Walnuts	590.1 mg/100 g	[109]
Beverages		
Cognac	31–55 mg/L	[121]
Oak-age red wine	0.0094 mg/L	[122]
Whiskey	1.2 mg/L	[122]
Seeds		
Marionvblackberry	3200 mg/100 g	[123]
Red raspberries	870 mg/100 g	[123]
Black raspberries	670 mg/100 g	[123]
Evergreen blackberry	2100 mg/100 g	[123]
Boysenberries	3000 mg/100 g	[123]
Longan	160 mg/100 g	[124]
Mango	120 mg/100 g	[124]

^1^ byproduct after pomegranate juice processing.

**Table 9 antioxidants-09-00707-t009:** Possible action mechanisms of the Type I antioxidants and related equations.

Action Mechanism	Chemical Equation	Features	Natural Compounds [128]
Type I			
HAT	H_n_Antiox + ^•^R → H_n−1_Antiox^•^ + HR	A key reaction mechanism	Polyphenols EA
PCET	H_n_Antiox + ^•^R → H_n−1_Antiox^•^ + H^+^ + ^•^ → HR	Exactly the same products as HAT	Flavonoids Quinone-hydroquinone
RAF	H_n_Antiox + ^•^R → [H_n_Antiox-R]^•^	Presence of multiple bonds peculiar of electrophilic radicals	Carotenoids Gentisic acid Rebamipide Hydroxybenzyl alcohols
SET	H_n_Antiox + ^•^R → H_n_Antiox^+^^•^ + R^−^	Primary pathway	EA Curcumin Carotenoids Catechins Edaravone Resveratrol
	H_n_Antiox + ^•^R → H_n_Antiox^+ −^^•^ + R^+^	Secondary pathway	Xanthones Carotenoids Trolox Caffeic acid Genistein
SPLET	H_n_Antiox → H_n__−1_Antiox^−^ + H^+^ H_n−1_Antiox^−^ + ^•^R → H_n__−1_Antiox^•^ + R^−^	Crucial mechanism in the scavenging activity in polar environments	Trolox Curcumin Vitamin E Quercetin Epicatechin Piceatannol Resveratrol Kaempferol Esculetin Fraxetin Morin Hydroxybenzoic Dihydroxybenzoic Flavonoids Isoflavonoids Xanthones Procyanidins Edaravone GA Erodiol
SEPT	(1) H_n_Antiox + ^•^R → H_n__−1_Antiox^•^^+^ + R^−^ (2) H_n−1_Antiox^•^^+^ → H_n__−1_Antiox^•^ + H^+^	A two-step mechanism involving electron transfer and deprotonation as in SPLET but in a different order	Vitamin E Galvinoxyl α-tocopherol Baicalein Astaxanthin Quercetin
SPLHAT	(1) H_n_Antiox → H_n__−1_Antiox^−^ + H^+^ (2) H_n−1_Antiox^−^ + ^•^R → H_n__−2_Antiox^•^^−^ + HR	Deprotonation of the antioxidant and an H transfer reaction	EA Anthocyanidins GA Esculetin α-Mangostin Propyl gallate

**Table 10 antioxidants-09-00707-t010:** Physicochemical computed properties of urolithin (URO) A and URO B.

Property Name	URO A	URO B
Molecular weight	228.2 g/mol	212.2 g/mol
XLogP3-AA	2.3	2.7
Hydrogen bond donor count	2	1
Hydrogen bond acceptor count	4	3
Rotatable bond count	0	0
Exact mass	228.0 g/mol	212.0 g/mol
Monoisotopic mass	228.0 g/mol	212.0 g/mol
Topological polar surface area	66.8 Å²	46.5 Å²
Heavy atom count	17	16
Formal charge	0	0
Complexity	317	289
Isotope atom amount	0	0
Defined atom stereocenter count	0	0
Undefined atom stereocenter count	0	0
Defined bond stereocenter count	0	0
Undefined bond stereocenter count	0	0
Covalently bonded unit count	1	1
Compound is canonicalized	Yes	Yes

**Table 11 antioxidants-09-00707-t011:** Production of UROs in different mammalian species.

Mammalian	Source	URO Type
Rat (*Rattus norvegicus*) ^1^	Pomegranate husk ^1^	A, B, C ^1^
Rat (*Rattus norvegicus*) ^1^	Ellagic acid ^1^	A ^1^
Rat (*Rattus norvegicus*) ^1^	Oak-flavored milk ^1^	A, B, C ^1^
Rat (*Rattus norvegicus*) ^1^	Pomegranate extract ^1^	A, M-6, M-7 ^1^
Rat (*Rattus norvegicus*) ^1^	Geraniin (*Geranium thunbergii*) ^1^	M-5 ^1^
Mouse (*Mus musculos*) ^1^	Pomegranate extract ^1^	A ^1^
Mouse (*Mus musculos*) ^1^	Pomegranate husk ^1^	A ^1^
Baver (*Castor canadensis*) ^1^	Wood ^1^	A, B ^1^
Complex toothed squirrel (*Trogopterus xanthipes*) ^1^	Unknown ^1^	A ^1^
Sheep (*Ovis Aries*) ^1^	*Trifoleum Subterraneum* ^1^	A, B ^1^
Sheep (*Ovis Aries*)	Quebracho ^1^	A ^1^
Cattle (*Bos primigenius*) ^1^	Young oak leaves ^1^	A, Iso A, B
Pig (*Sus scrofa domesticus*) ^1^	Acorns ^1^	A, C, D, B
Humans (*Homo Sapiens*) ^1^	Pomegranate juice ^1^	A, C, Iso A, B
Humans (*Homo Sapiens*) ^1^	Pomegranate extract ^1^	A, B, C
Humans (*Homo Sapiens*) ^1^	Walnuts ^1^	A, B, C
Humans (*Homo Sapiens*) ^1^	Strawberry ^1^	A, C, Iso A, B
Humans (*Homo Sapiens*) ^1^	Raspberry ^1^	A, C, Iso A, B
Humans (*Homo Sapiens*) ^2^	Blackberry ^2^	A, C ^2^
Humans (*Homo Sapiens*) ^3^	Cloudberry ^3^	A ^3^
Humans (*Homo Sapiens*) ^1^	Oak-aged red wine ^1^	A ^1^
Humans (*Homo Sapiens*) ^1^	Tea ^1^	A ^1^
Humans (*Homo Sapiens*) ^1^	Nuts ^1^	A, Iso A, B ^1^

^1^ [38]; ^2^ [146]; ^3^ [44].

**Table 12 antioxidants-09-00707-t012:** Animal and human biological fluids and tissues where UROs and/or their conjugates and/or nusatins were detected in several experimental types and conditions and after intake of different ET-rich foods [43].

Models N° Tests for Animal Type	Biological Fluid	Tissue	UROs Type in Biological Fluids	UROs Type in Tissues
Animals ^1^ Tests = 20	Urine	Kidney	A, A-glur, A-3-glur, B, C, B-sulfate, B-glur, C, D-derivatives, Iso A-3-glur, Iso A-9-glur, Iso B-glur, A-sulfate, Iso A-sulfate, A-glur-sulfate, Iso A-sulfo-glur, A-diglur, Iso A-diglur, nasutin-A-glur, Isonasutin-A-glur, trihydroxy-urolithin-diglur, dihydroxy-urolithin derivative	M-7, A-glur, A, B, A-sulfate
	Feces	Liver	A, Iso A, B, M-6, C, Nasutin,	A, M7, A-glur, A-sulfate,
	Plasma	Intestinal lumen	A, B, C, 8-methy-C, C, D-derivatives, A-glur, B-glur, nasutin-A-glur, trihydroxy-urolithin-diglur, trihydroxy-urolithin derivative, A-sulfate, Iso A-glur, Iso A-sulfate, B-sulfate, Iso A-solfo-glur	A, C, D, A-sulfate
	Blood	Intestinal tissue		A, C, D, A-glur, A-sulfate
	Bile	Hurt	A, C, D, A-glur, nasutin-A-glur, Isonasutin-A-glur	B
	Ruminal fluid	Brain	A, Iso A, B, C, A-glur.	A, Methyl-A
	Beaver excretions	Lung	nasutin-A-glur	A, A-sulfate, A-glur,
		Prostate		A, A-sulfate, Methyl-A, A-glur
		Colon		A, A-sulfate, Methyl-A, A-glur
		Cecum		A, Nasutin
		Stomach		Nasutin
Humans ^2^ Tests = 27 ^3^	Urine	Prostate	A, B, C, M-5, D, B-sulfate, A-glur, Iso A, Iso A-glur, B-glur, A-3-glur, A-8-glur, trihydroxyurolithin, A-sulfate, Methyl-A, A-sulfo-glur, D-methyletherglur, A-diglur, Dimethy-C, C-glur, D-glur, C-methylethersulfoglur, Methyl-C-glur, Methyl-D-glur, C-sulfate,	A, A-glur, B-glur,
	Feces	Normal and malignant colon tissue	A, B, C, D, E, M-5, M-6, M-7, Iso A	A-glur, D, Iso A-glur, A-sulfate, C, B-glur, A, Iso A, B-sulfate, B, M-5, M-6
	Plasma		A, B, D, A-sulfate, B-sulfate, trihydroxyurolithin, A-glur, Iso A-glur, B-glur, C, Methyl-A, C-glur, Methyl-C-methyletherglur, C-diglur, D-glur, A-sulfo-glur, C-sulfate, IsoA-sulfate,	

^1^ Rats, Iberian pigs, bull, sheep, mice, greenfinches, beetle, beaver, squirrel, pig; ^2^ Healthy or non- healthy volunteers, patients with COPD, ileostomy, PBH, PCa, MetS, CRC; ^3^ Number of replicates for test. Abbreviations: COPD, chronic obstructive pulmonary disease; PBH, benign prostatic hyperplasia; PCa, prostate cancer; MetS, metastatic disease; CRC, colorectal cancer or colorectal carcinoma.

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
