# Peer review of "Oxidative Stress, Antioxidant Capabilities, and Bioavailability: Ellagic Acid or Urolithins?"

_antioxidants, 2020, doi:10.3390/antiox9080707_

Round 1

Reviewer 1 Report

The authors prepared good review of the knowledge on  oxidative stress in relation to ellagic acid and urolithins. 

I suggest the correction

table 3 radical instead specie

table 5 in horizontal page layout

line 558 please delete three points

Table 11 Authors should explain or correct "Animals1/20" and "Human2/27/3"

Author Response

The authors prepared good review of the knowledge on oxidative stress in relation to ellagic acid and urolithins.

I suggest the correction

table 3 radical instead specie

As suggested by the Reviewer, in the title of Table 3 (line 244), in the title of column 1 and in the text (line 242), “species” has been replaced by “radicals”. Anyway, since the Table contains also the chemical specie ONOOCO2, which is not a radical, it has been indicated as reactive specie and specified both in the text and in the Table and Table titles.

table 5 in horizontal page layout

As requested by the Reviewer, Table 5 layout has been set up in horizontal. Please see from line 280.

line 558 please delete three points

The three point has been deleted. Please see line 581 of revised version of the manuscript.

Table 11 Authors should explain or correct "Animals1/20" and "Human2/27/3"

As requested from the Reviewer, column 1 of the original Table 11, now Table 12, containing the unclear wordings indicated, has been modified. In addition, footnote 3 has been better expressed. Please see lines 630-631.

Reviewer 2 Report

The manuscript entitled " Oxidative Stress, Antioxidant Capabilities and Bioavailability: Ellagic acid or Urolithins?" has novel information.

Figure 3a looks blurred, please change it.

There are no more comments for this review.

Author Response

The manuscript entitled " Oxidative Stress, Antioxidant Capabilities and Bioavailability: Ellagic acid or Urolithins?" has novel information.

Figure 3a looks blurred, please change it.

The authors agree with the Reviewer. In order to improve Figure 3a, all the Figure 3 has been remade.

There are no more comments for this review.

Reviewer 3 Report

The review paper deals with the properties of selected natural polyphenols for counteracting oxidative stress, triggered by overproduction of reactive oxygen and nitrogen species, which is the main responsible of several human diseases. Among them, ellagitannins, ellagic acid and its metabolites urolithins drew a particular attention. In particular urolithins demonstrated in vivo the ability to reach tissues in a greater extent.

The review is well-written, logically-structured and with a satisfactory number of references. Also a discussion on the clinical applicability of urolithins over ellagic acid was also include. On the basis of such considerations I suggest publication in the present form.

Author Response

The review paper deals with the properties of selected natural polyphenols for counteracting oxidative stress, triggered by overproduction of reactive oxygen and nitrogen species, which is the main responsible of several human diseases. Among them, ellagitannins, ellagic acid and its metabolites urolithins drew a particular attention. In particular urolithins demonstrated in vivo the ability to reach tissues in a greater extent.

The review is well-written, logically-structured and with a satisfactory number of references. Also a discussion on the clinical applicability of urolithins over ellagic acid was also include. On the basis of such considerations I suggest publication in the present form.

The authors thank the Reviewer for its comment.

Reviewer 4 Report

This review article written by Silvana Alfei and coauthors described antioxidant capabilities and bioavailability of two chemicals ellagic acid and urolithins. The authors spent a lot of efforts to prepare this review and reported the chemical properties and bioactivities of EA and UROs. However, some content of this review was repeated or well-known information. I think the current content of this review is not really suitable to publish in Antioxidants.

The comments are list below:

  1. In section 2 Oxidative Stress, I feel it was mention in several papers. I think the authors should delete or shorten this part. Especially some citing papers are a textbook or a review (refs 48, 49, and so on).
  2. Line 468, figure 2, I cannot find the structure of UROs A and B and IsoUROs A and B.
  3. For 6.1 structure and chemistry of UROs, I think the authors can provide the structure and propriety of URO-A, iso
  4. Table 10, please remove the column of animal pictures. Since the scientific name was presented. Where are the *unpublished results? In table 10, I cannot find the aster.
  5. Line 503, O-desmethylangolensin, the O should be italic.
  6. Lines 539-544, those sentences are irrelevant to the topic of this review. Please delete those sentences.
  7. Comparing to EA, the mechanisms of the antioxidant effects of UROs are relatively less.

Author Response

This review article written by Silvana Alfei and coauthors described antioxidant capabilities and bioavailability of two chemicals ellagic acid and urolithins. The authors spent a lot of efforts to prepare this review and reported the chemical properties and bioactivities of EA and UROs. However, some content of this review was repeated or well-known information. I think the current content of this review is not really suitable to publish in Antioxidants.

The comments are list below:

In section 2 Oxidative Stress, I feel it was mention in several papers. I think the authors should delete or shorten this part. Especially some citing papers are a textbook or a review (refs 48, 49, and so on).

The authors agree with the Reviewer. The part concerning OS in Section 2 has been reduced and modified and some references have been removed. Please see lines 137-144, 149-153, 160-162, 179-192, 238-241, 261-265 and 271-272.

Line 468, figure 2, I cannot find the structure of UROs A and B and IsoUROs A and B.

The authors apologize to the Reviewer for their distraction. In the line 468 of the unrevised manuscript (now line 485) Figure 2 is not correct. Figure 4 is correct. The authors thank the Reviewer for having reported such error. The section in which the Figure is observable has been included. Please see lines 485 and 496-497.

For 6.1 structure and chemistry of UROs, I think the authors can provide the structure and propriety of URO-A, iso

Depending on the typing error explained in the previous point, the Reviewer could not observe the UROs structures which he requires, actually already included in the unrevised manuscript and available in Figure 4 in Section 3.2., where the EA metabolism is discussed. The authors consider redundant to report UROs structures again, and retain sufficient refer to the previous Figure 4.

Concerning to provide UROs properties, the authors agree with the Reviewer. In order to satisfy this request a new Table (Table 10, line 488) reporting the computed physicochemical properties of URO A and URO B has been included in Section 6.1.

Table 10, please remove the column of animal pictures. Since the scientific name was presented. Where are the *unpublished results? In table 10, I cannot find the aster.

The column of animal pictures has been removed by the original Table 10, now Table 11. *unpublished results is a mistake and it has been removed. Please see line 573.

Line 503, O-desmethylangolensin, the O should be italic.

“O” has been written in italic. Please see line 524 of the revised manuscript.

Lines 539-544, those sentences are irrelevant to the topic of this review. Please delete those sentences.

As requested from the Reviewer, lines 539-544 of the unrevised manuscript have been deleted. Please see lines 560-565 of the revised paper. In addition, the text in lines 566-571 has been modified.

Comparing to EA, the mechanisms of the antioxidant effects of UROs are relatively less.

Concerning this last comment from the Reviewer, the authors are not sure to understand what the Reviewer means. No request is present in this point, but only a comment which simply seems a consideration of the Reviewer.

Round 2

Reviewer 4 Report

The authors have already corrected their manuscript according to my suggestion. I also found that other reviewers seem to appreciate this work and want to accept this paper in the present form. For these reasons, I feel this paper can be accepted and published in Antioxidants.